# Sylvatic Canine Morbillivirus in Captive *Panthera* Highlights Viral Promiscuity and the Need for Better Prevention Strategies

**DOI:** 10.3390/pathogens10050544

**Published:** 2021-04-30

**Authors:** Mainity Batista Linhares, Herbert E. Whiteley, Jonathan P. Samuelson, Shih Hsuan Hsiao, Adam W. Stern, Ian T. Sprandel, Patrick J. Roady, David A. Coleman, Rebecca Rizzo, S. Fred Froderman, Karen A. Terio

**Affiliations:** 1School of Veterinary Science, The University of Queensland, Gatton, QLD 4343, Australia; 2Zoological Pathology Program, University of Illinois, c/o Chicago Zoological Society 3300 Golf Road, Brookfield, IL 60513, USA; kterio@illinois.edu; 3Veterinary Diagnostic Laboratory, University of Illinois at Urbana-Champaign, 1224 Veterinary Medicine Basic Sciences Building, 2001 South Lincoln Avenue, Urbana, IL 61802, USA; hwhitele@illinois.edu (H.E.W.); jpsamue2@illinois.edu (J.P.S.); shsiao1@illinois.edu (S.H.H.); adamstern@ufl.edu (A.W.S.); isprandl@illinois.edu (I.T.S.); roady@illinois.edu (P.J.R.); 4College of Veterinary Medicine, University of Florida, 2015 SW 16th Avenue, Gainesville, FL 32608, USA; 5The Jackson Laboratory, 600 Main Street, Bar Harbor, ME 04609, USA; david.coleman@jax.org; 6Exotic Feline Rescue Center, 2221 E Ashboro Road, Center Point, IN 47840, USA; rizzor81@yahoo.com; 7White Stone Veterinary Clinic, 5423 Calvert Ln., Plainfield, IN 46168, USA; frodersf@icloud.com

**Keywords:** canine distemper virus, spillover, outbreak, pneumonia, vaccine, Panthera, tiger, lion

## Abstract

Canine Distemper Virus (CDV) is a multi-host morbillivirus that infects virtually all *Carnivora* and a few non-human primates. Here we describe a CDV outbreak in an exotic felid rescue center that led to the death of eight felids in the genus *Panthera*. Similar to domestic dogs and in contrast to previously described CDV cases in *Panthera*, severe pneumonia was the primary lesion and no viral antigens or CDV-like lesions were detected in the central nervous system. Four tigers succumbed to opportunistic infections. Viral hemagglutinin (H)-gene sequence was up to 99% similar to strains circulating contemporaneously in regional wildlife. CDV lesions in raccoons and skunk were primarily encephalitis. A few affected felids had at least one previous vaccination for CDV, while most felids at the center were vaccinated during the outbreak. *Panthera* sharing a fence or enclosure with infected conspecifics had significantly higher chances of getting sick or dying, suggesting tiger-tiger spread was more likely than recurrent spillover. Prior vaccination was incomplete and likely not protective. This outbreak highlights the need for further understanding of CDV epidemiology for species conservation and public health.

## 1. Introduction

Canine distemper virus (CDV) in the genus *Morbillivirus*, family *Paramyxoviridae*, can naturally infect all families of terrestrial carnivores as well as rhesus and cynomologus macaques [1,2,3]. Worldwide, CDV has been reported in wildlife, including numerous endangered species [4], and it raises conservation concerns for species such as Amur tigers, lions, cheetahs and African wild dogs, among others [5,6,7,8,9,10,11,12].

CDV infection occurs primarily through the respiratory epithelium and mucosal associated leukocytes, in particular macrophages and dendritic cells [5,13]. Similar to other morbilliviruses, CDV utilizes the signaling lymphocyte activation molecule (SLAM) to infect leukocytes and then multiply and spread systemically, particularly through the immune system. Approximately 10 days after initial infection, secondary viremia is associated with viral inclusion bodies in virtually all epithelial cells in the body and polioencephalitis may develop [13,14]. At this point, nectin 4-positive cells are thought to be the primary target and considered a key receptor for epithelial infection, cell-cell spread, syncytial formation and virus shedding [15]. Domestic dogs that develop mild immune reaction or are infected with moderately neuropathogenic strains, may survive the two first viremia waves and develop a late (approximately three weeks later or more, and typically demyelinating) neuropathological condition that can also issue severe neurological signs/deficits [5,13,14,16]. Thus, clinically, animals infected with CDV may present with respiratory, gastrointestinal and/or central neurological signs [1,5,17,18]. Morbidity and mortality vary greatly within species [1,19] as it is dependent on the viral strain and host factors such as immunocompetence [1,20]. Secondary infections with bacteria, protozoa and/or fungi are common and may complicate the clinical course or be ultimate cause of demise [1,21]. Secondary infections are most likely a consequence of severe immunosuppression elicited by lymphocytolysis of both T and B-cells among other mechanisms [13]. Previous descriptions of CDV infection of non-domestic felids of the genus *Panthera,* have documented central nervous signs such as Grand mal seizures and encephalitis [1,5,11,22].

The hemagglutinin (H) gene is frequently sequenced and analyzed for epidemiological studies of CDV [23,24]. The H-protein binds to both SLAM and nectin 4 host-receptors at two non-overlapping sites [15] and is among the most variable genes of CDV [24,25]. Residues 519, 530 and 549 are within the SLAM binding region and considered to be under significant positive selective pressure [26,27]. All these sites have been subject of speculation regarding the CDV’s ability to jump species, particularly to non-canids [27,28,29,30]. However, an ever growing number of authors see no significant difference among residues in multi-host strains or virus-targeted host receptors [19,26,31,32,33], but rather, pleiotropic and temporal, sometimes concurrent, geographical clades seem to be the cause for outbreaks and CDV evolution [23,33,34,35]. Nonetheless, though H-gene has been largely used for phylogenetic studies, full genome phylogenetic trees are increasingly preferred and have helped identify other genetic sites where recombination may play a significant role in CDV evolution [24,36]. Further, these studies have showed that despite relatively high recombination rate for a morbillivirus, CDV genes are largely under negative selection [24], its evolution is predominated by homologous recombination and residue 178 of H-gene is one of the nine positive selection sites and involved in immune escape [36].

At this time, vaccination of exotic felids is extra-label [37] and efforts have been made to assess the safety and effectivity of both recombinant canarypox-vectored CDV and modified live CDV vaccines in felids [38]. Overall, CDV vaccination of felids in captivity is recommended only if animals are considered at high risk and usage of modified live CDV vaccines is discouraged given the risk of inducing disease [32,39,40]. Captive felids in zoological or rescue facilities are considered at high risk if they live within areas where sylvatic species, e.g., raccoons, serve as a reservoir and may thus be a vector of the virus [18,22,41]. Here, we describe the pathology, and epidemiology of a CDV outbreak in a captive population of felids of genus *Panthera*. Diseased *Panthera* primarily developed pneumonia and had a relatively long survival time with different opportunistic secondary infections. Local, free-ranging raccoons were deemed most likely source of virus as evidenced by similarity of H-gene sequences isolated from submitted cases. Key amino acid residues (178, 519, 530 and 549) from the H-gene protein were compared. The same virus strain/clade is believed to have spread within the raccoon metapopulation spanning two states and to have caused death of a captive snow leopard in a neighbor state, shortly after the outbreak recorded in the rescue center [22].

## 2. Results

### 2.1. Clinical Signs, Histopathology and Molecular Testing

From a population of 187 non-domestic felids, over a two-month period, 10 non-domestic felids of the genus *Panthera* were submitted for necropsy. Of these 10 cases, 8 animals presented with a common clinical history of mild to severe respiratory signs and were diagnosed with canine distemper virus infection. The index case of this outbreak was one of two tigers submitted for necropsy on 26th of October of 2015 (Table 1). Time from initial signs until death in post-mortem CDV-confirmed cases ranged from less than 1 to 40 days and these animals were 5 to 20 years (mean 15 years) old.

At necropsy, seven *Panthera* had pneumonia ranging from mild interstitial to severe bronchopneumonia. Eight animals had histopathological signs consistent with interstitial to bronchointerstitial pneumonia, which were associated with intracytoplasmic and intranuclear eosinophilic inclusion bodies within syncytial cells. Type II pneumocyte hyperplasia and secondary infections were also commonly observed (Figure 1 and Figure 2). Characteristic viral inclusions were noted in pulmonary (8/8), urinary (2/8), gastric (2/8), pancreatic (2/8), duodenal (1/8) and adrenal gland (1/8) epithelium. CDV infection was confirmed by immunohistochemistry (Figure 1) and RT-PCR in the lung of all eight *Panthera*. Infection with CDV was further confirmed by fluorescent antibody (FA) testing of fresh lung tissue in one tiger (Appendix A) and through virus isolation in two tigers and one lion (Table 2). Secondary infections were observed in three tigers (case numbers 5, 6 and 7), which had the longest disease course (see Table 1). One tiger (case 5) had a secondary bacterial pneumonia from which *Enteroccoccus faecalis*, *Escherechia coli* and *Proteous mirabilis* were cultured. Another tiger (case 6) had protozoal pneumonia and lymphadenitis with intralesional *Toxoplasma gondii* confirmed by immunohistochemistry, and the third tiger (case 7) had systemic *Histoplasma capsulatum* infection (Figure 2). Other findings were considered age-related degenerative lesions and neoplastic lesions common in these species.

During the outbreak and shortly thereafter, regional raccoons (three from Indiana and one from Illinois) and two skunks from Illinois were necropsied. One raccoon from IN was negative and remaining skunks and raccoons were proven to be infected with CDV through IHC (4/5) and/ or RT-PCR (4/5) with subsequent H-gene sequencing (4/5). Interestingly, these animals had minimal to no pneumonia and primary acute encephalitis (4/5) with intralesional CDV particles (Appendix A). The one raccoon captured approximately 3.3 km from the center during the outbreak had generalized lymphadenomegaly grossly, which was associated with moderate lymphoid depletion histologically. CDV-FA in tissue homogenates from lung and brain were negative but CDV-infection was confirmed both via RT-PCR and H-gene sequencing.

### 2.2. Epidemiologic Study

The rescue center was characterized as two distinct epidemiologic units, unit A and B (approximately 640 m apart, see Figure 3 and Table 3). Within unit A, enclosures frequently shared fences (N = 12/63) and group size ranged from 2 (N = 7), 3 (N = 3), 4 (N = 2), 5 (N = 1) to 7 felids (N = 1). In unit B, enclosures (N = 77) were slightly more widespread with only two enclosures that shared one fence. Groups sizes in unit B ranged from 2 (N = 16), 3 (N = 5), 4 (N = 2) to 8 animals (N = 2). Groups in unit A and B were of mixed species, frequently cohousing cougars, lions, tigers and/or leopards. Staff and cleaning utensils are shared between all enclosures and units. The majority of CDV-dead *Panthera* (N = 6/8) were housed in groups or enclosure with shared fences. Thus, felids held in enclosures with shared fences were defined as metagroups. Though the majority of felids were housed alone (N = 35/68 in unit A and 27/100 in unit B), shared fences in unit A led to metagroups of two or five felids at time, in which four *Panthera* died of CDV and secondary infections. In unit B, the only tiger to die was housed in an enclosure that shared fences and was thus considered to be in a metagroup of three tigers. Thus, shared fence and or enclosure with a sick conspecific were considered a risk factor.

Unit A housed 85 total felids of which 68 were deemed CDV-susceptible (Table 3). Unit A was the epicenter of the outbreak with seven of the eight total CDV-confirmed deaths (10.29%, 7/68), while Unit B housed 100 CDV-susceptible (of 103 total) felids with only one mortality (1%). Multiple felids were reported ill with consistent clinical signs but for whom infection could not be confirmed including 23 animals in Unit A (33.8% morbidity, *N* = 68) and 2 animals in Unit B (2% morbidity, *N* = 100). When sharing fences or enclosure with a CDV-confirmed diseased conspecific, the odds ratio for *Panthera* developing signs of illness (sick OR) or dying of CDV-infection (Death OR) were 75 (*p*-value < 0.0001) and 9.44 (*p*-value = 0.0043), respectively. These findings suggest spread among *Panthera* after initial spillover from wildlife. Further, the differences in mortality and morbidity between Unit A and B were significant (*p*-values = 0.0079 and <0.00001, respectively), and suggest that other ubiquitous risk factors such as fomites, i.e., shared cleaning/feeding tools and staff members, were of minimal significance for viral spread in the facility. Nonetheless, spacing between enclosures in Unit B was better and seem to have been protective to that portion of the population.

Since 2004, the ability of the rescue center to vaccinate felids has varied greatly, with a varying number of felids vaccinated every two to three years. Around time of rescue, all felids were reported to receive one dose of Fel-O-Vax IV + CalciVax (Elanco US, inactivated viruses and bacteria vaccine that covers feline Rhinotracheitis, Calicivirus, Panleukopenia virus and Chlamydia Psittaci); lions, tigers and leopards were also vaccinated with one dose of Nobivac ^®^ Puppy-DPv (Merck Animal Health, modified live vaccine that covers canine distemper and parvoviruses). There were no reports of adverse reactions to or development of vaccine-induced disease after vaccination with either vaccines. At the time of the outbreak, from a total 168 CDV-susceptible felids, only 33 animals had available proof of one previous CDV-vaccination (5 at Unit A and 28 at Unit B). Vaccinations were given within one year the animals’ arrival to the facility for the majority of previously vaccinated *Panthera* (*N* = 21) and fewer animals (*N* = 12) had records of one CDV-vaccine between 2 and 10 years after their arrival. Before 2004, Felids would receive at least one dose of either the PUREVAX ^®^ Ferret Distemper (Merial, recombinant canarypox vector expressing the HA and F glycoproteins of canine distemper virus) or Nobivac ^®^ Puppy-DPv. Since 2004 and based on available records, at least 146 animals had been vaccinated with Nobivac ^®^ Puppy-DPv. During the outbreak, most felids (*N* = 145) were vaccinated one to two times (between end of September and end December) with Nobivac ^®^ Puppy-DPv. The vaccinations did not follow a systematic immunization schedule (see Appendix A).

Of the five vaccinated (Nobivac ^®^ Puppy-DPv) *Panthera* in unit A, one showed clinical signs and died of CDV (case # 4). All other deaths had no records of CDV-vaccination (six animals from unit A and one from Unit B) and in unit B only two animals showed clinical signs of disease during the outbreak. One of these two cases was the last CDV-confirmed case and the second was a non-CDV related death characterized by diffuse bacterial enteritis, mixed bacterial pneumonia and meningitis (mixed infection with *Enterococcus faecalis, Klebsiella pneumoniae* and *Streptococcus zooepidemicus*). The latter had been vaccinated during the outbreak and tested negative for CDV in all of the following diagnostic attempts: immunohistochemistry (brain), direct immunofluorescence assay (lung) and virus isolation (lung). This last death occurred 32 days after onset of clinical signs, 47 days after the index case of the outbreak in the center, and 42 days after its vaccination. Given the success rate of the different diagnostic tests described in the eight Panthera and five wildlife cases herein, we believe that had this animal been infected with CDV: (1) viral particles would had been isolated or detected through direct FA (particularly during secondary viremia with presence of prolific viral particles in all epithelial cells); (2) associated histological changes would have been identified (acute or chronic, particularly given that all pathologists were now highly sensitive for changes compatible with CDV and other similarly chronic cases were identified); (3) immunohistochemistry assay, which had been previously validated in lions and worked in multiple sections of lungs from the *Panthera* and brain from wildlife herein described, would have been positive in at least one section of submitted tissue. Thus, we believe that it was very unlikely this animal was misdiagnosed as CDV-negative.

Lastly, all felids or group of felids that showed any type of clinical signs were reported to be treated with broad spectrum antibiotic when fed. Felids that did not respond to group therapy were taken to another, more isolated area with fewer enclosures to be more closely monitored and treated. This was the case for two CDV-cases treated the longest (cases #6 and 7) as well as the one CDV-negative tiger mentioned above. These animals were kept separately.

### 2.3. H-Gene Sequence Analysis

H-gene sequences from all *Panthera* and raccoon submitted from an adjacent wildlife area had 99–100% similarity. Sequences were also available from two skunks and one raccoon from a neighboring state (Illinois, IL, USA, approximately 300 km north from the rescue center in Indiana, IN, USA) collected shortly after the outbreak at the feline rescue center. Of these animals, the H-gene sequences of the skunk and one raccoon, were respectively 96.6% and 97% similar to the H-gene sequence detected in the *Panthera* and raccoon. Additionally, sequence and amino acids at residues 178, 519, 530 and 549 were also similar (Table 4, sequences were deposited with GenBank, accession numbers are MW984525 to MW984536). These findings are suggestive that the outbreak experienced in the *Panthera* of the rescue center in Indiana was mostly likely due to a CDV-strain shared with the local raccoon population. Further, the CDV-sequence detected in the lion was 96% similar to a sequence harvested from a fatal infection in a snow leopard held in captivity (internal report and [22]).

## 3. Discussion

This report describes an unusual outbreak of CDV in captive *Panthera* and sympatric wildlife. Affected animals were infected with similar viral strains (based on H-gene sequences) to those noted in regional wildlife suggesting that they were part of one continuous outbreak. In all, the outbreak affected animals of at least five different species and in at least two states. There was minimal variation among CDV in these cases at both the genetic, amino acid (particularly residues 530 and 549) or associated-disease level. Further, epidemiological studies suggested that interspecies infection and spread of virus among *Panthera* was frequent.

Although reports of CDV in the genus *Panthera* date back to 1980s [42], clinical signs and course of disease in felids in the wild or in captivity are scarce. In previously described cases, disease in *Panthera* is mainly associated with severe central neurological signs [10,17,22,42,43]. However, as noted in our case series as well as by Appel et al., 1994, CDV-infected *Panthera* can show a range of subtle and non-specific clinical signs varying from mild serous or mucopurulent nasal discharge and non-specific behavior changes, to mild diarrhea and occasional sudden onset of marked respiratory signs. This is in line with canine distemper in dogs and other species [1,44] and it is important to communicate to both veterinarians and non-veterinarians who work with *Panthera* that infection with CDV should be included as a differential when animals are presented with non-neurologic clinical signs, especially when located in areas where CDV is endemic or has been recently introduced. It is also noteworthy that *Panthera* in this outbreak survived as long as 40 days and three of them succumbed to opportunistic infections, which could mask or preclude detection of viral particles and/or targeted diagnostic approach.

Comparison of the H gene of CDV by some researchers has suggested that strains with 549H are more generalist pathogens and more likely to occur in multiple hosts, particularly non-canids [26,27,29,45]. In this series, the *Panthera* (tigers and lion), *Procyonids* (raccoons) and *Mephitidae* (skunk) all had 549H (base sequence CAC). Additionally, residue 178, which has been associated with viral evasion of immune response and is considered under high positive selection pressure [36], was also preserved (base sequence GGC, 178G) throughout the samples. Whether this suggests a novel species-jump or simply spillover of the strain circulating temporally and geographically is uncertain. The latter, however, is likely and in line with other reports [4,10,17,18,22,35,41,43,46,47]. Additionally, the geographical spread between states and from sylvatic to urban areas, as noted in this case, is no novelty and relies on co-existence of susceptible species (domestic, non-domestic, free-ranging or in managed care). Notably, during this larger outbreak, a raccoon and a skunk from Illinois were minimally (around 3–4%) different from the H-gene detected in the cluster from Indiana. Although residues 530, 549 and 178 were conserved, with a homologous recombination at 530 (GGT/GGC, 530G), recombination at residue 519 was not (AGA-R to ATA-I). On one hand, the homologous difference at residue 530 is in line with recent evolution studies on CDV [36]. On the other hand, R519I and 549H have been associated with disease occurrence in *Panthera* and in association with Grand mal seizures (e.g., Serengeti outbreak [30]). The latter might explain, why the skunk and the raccoon, had lesions primarily on the brain and lesser in the lung, whereas no central nervous signs were reported and no histopathological changes compatible with CDV-infection were noted in *Panthera* during the outbreak. It is also noteworthy that a snow leopard that succumbed to CDV infection around the time of the outbreak in Indiana, not only had both pulmonary and cerebral changes compatible with CDV, but also presented with primary neurological signs (seizures, [22]). Full genome sequence from virus isolates between these incidents, might help elucidate further, whether recombinant evolution played a significant role in the striking different clinical course between the *Panthera* in Indiana, and the raccoon, skunks and snow leopard in Illinois. Other elements that likely also contributed to different clinical outcomes were infection pressure and host immune response (including a potential protective role of the off-label vaccination).

Morbidity and mortality varied significantly within the areas of the rescue center, with the most densely populated area, Unit A, being the epicenter of the outbreak (7/8 deaths). Sharing enclosure and or fence with sick conspecifics was a significant risk factor, evidencing interspecies spread among *Panthera* following spillover from wildlife. Along with well-conserved H-genes shared with local wildlife, this outbreak further supports the argument that no significant species-adaptation is needed for CDV species-jump and spread. In addition to other natural spillovers [3,10,17,33,48], experimental reports have shown that the same holds true even for cells expressing human nectin-4 [49]. Bieringer et al. 2013 also reported that a single functional adaptation was required (D540G) for CDV adaptation to human SLAM. All strains detected in this outbreak had 540D with conserved base pair sequence (data not shown). Nonetheless, other mutations discussed above and genetic variance exceeding 3% in the H-genes sequences detected could be reflective of growing scientific evidence that CDV has a high mutation rate and that coexistence of clades promote evolution [24,29,36]. Conversely, given the proximity of wildlife to the rescue center and the fact that the center has existed for over a decade, it is puzzling that *Panthera* only showed fatal-CDV infection now. Retrospective and prospective serological studies might help identify subclinical infections, as noted in similar studies in lions from Serengeti [30,42]. Further, little is known about the epidemiological role of species that seroconvert but seem to not develop disease, such as cougars, ocelot and the domestic cat [17,50,51]. In other words, contemporaneous and comprehensive surveillance covering multiple species is needed in order to further understand virus spread, clinical course and epidemiological role of the numerous co-existing species in a given geographic area. This shall allow a better understanding of risk factors promoting species-jump and host-adaptation of CDV [8], as well as point towards most effective wildlife and domestic animal health management strategies to decrease CDV spread and species-jump.

Although vaccination scheme and record keeping at the rescue center has varied through the years, it was reported that *Panthera* were systematically vaccinated around time of rescue with few available supporting records. From the 33 *Panthera* with confirming records of vaccination against CDV previous to the outbreak, only one showed clinical signs and died from CDV. Additionally, 24 of the reported sick and all other 7 deaths had no available records to confirm previous vaccination. Thus, we are limited in assessing the protective role of vaccination. Nonetheless, 145 animals were vaccinated one to two times during the outbreak, and no adverse reactions were reported, despite the off-label usage of modified live vaccine. In addition, one of the tigers necropsied during the outbreak, had been vaccinated shortly before dying but no CDV-particle was detected via immunohistochemistry, direct immunofluorescence or virus isolation. Altogether, this suggests that off-label usage of the live-modified Nobivac Puppy-DPv vaccine does not elicit disease or adverse reactions in *Panthera.* It remains uncertain whether the reported vaccination elicited an antibody response. In order to assess the latter, specific serological tests that distinguish between immune response to vaccination and natural infection would be necessary. Unfortunately, such serological tests are rarely available in both human and veterinary medicine. Additionally, similar to currently available vaccines, serology tests differentiating between vaccinated and naturally infected *Panthera*, would also be off-label or require additional validation previous to application. Alternatively, screening of banked sera for neutralizing antibodies could be attempted. However, rescue centers as well as wildlife monitoring programs frequently rely on very limited resources, and, thus, have limited numbers of samples stored or the capability to do so. Furthermore, many large carnivores require sedation or anesthesia for simple medical procedures (e.g., blood collection), which only increases the risks and costs associated with the intervention. Lastly, given that the animals were kept within a region that CDV was endemic, it would be difficult to select a previous timepoint when the *Panthera* were certain to be seronegative for CDV, particularly with the potential of the animals having been vaccinated at the time of rescue.

Finally, this case report reminds us of the importance of comparative pathology and medicine, the complexity of the epidemiology of diseases shared with wildlife. In the face of the ever growing challenges regarding wildlife health and the implication of anthropogenic factors such as (1) encroaching into wildlife; (2) confinement or captivity of different non-domestic and domestic species under illegal (e.g., animal markets) or legal terms (e.g., rescue centers, zoos, urban areas in proximity to wildlife reservoirs); and (3) the growing controversy and public skepticism towards vaccination, it is urgent that more resources are allocated in surveilling and monitoring disease emergence and reemergence in wild- and domestic species as well as developing protective measures (including vaccines) that are safe and indeed protect susceptible species.

## 4. Materials and Methods

### 4.1. Case Series

Between October and December 2015, the Veterinary Diagnostic Laboratory at the University of Illinois at Urbana-Champaign performed diagnostic necropsy and histopathologic examination of eight tigers, one lion, one puma and one raccoon submitted from the same geographical location. Additionally, three raccoons and one skunk from the larger metapopulation in an adjoining state were also submitted. Two animals were humanely euthanized for welfare reasons and nine animals died naturally. Representative tissues from each submission were saved in 10% neutral buffered-formalin and processed and stained routinely for histopathology. Sections of lung from all CDV-confirmed cases were saved frozen (−80 °C), of which selected cases also had sections of brain saved frozen.

*Panthera* were from a population of 187 non-domestic cats at a single facility. From these 187 animals, 168 animals were from species in which CDV-disease and presence of virus material had been reported: *Lynx canadensis*, *Lynx rufus*, *Panthera leo, Panthera tigris, Panthera pardus* and *Panthera uncia* [4,17,52]. Felids kept in the rescue center frequently have unknown date of birth, thus time spent at the center was used to estimate their minimal age. At time of the outbreak, felids had been kept for a mean of 8 years. However, 4 tigers (estimated to be of 10 years of age) were introduced in September 2015 (preceding the outbreak) and 79 felids had been in the center for at least 10 years (with a mean of 13 years, and a maximum of 19 years).

### 4.2. Laboratory Investigation

Immunohistochemistry was performed using a commercially available mouse monoclonal anti-CDV nucleoprotein antibody (Custom Monoclonals International, Sacramento, CA, USA) following the manufacturer’s instructions [11]. Three-micrometer sections of formalin-fixed, paraffin-embedded blocks were cut and placed on positively charged slides (Probe-On Plus, Fisher Scientific, Springfield, NJ, Australia), deparaffinized and hydrated using a series of graded alcohols. Antigen retrieval was performed by immersing slides in citrate buffer (Unmasking solution, Vector Laboratories Inc., Burlingame, CA, USA) for 20 min at 90–95 °C. Treated slides were cooled at room temperature. Endogenous peroxidase was blocked with 0.3% H_2_O_2_. Nonspecific binding sites were blocked by application of 2% goat serum. Slides were then incubated with a commercially available mouse monoclonal anti-CDV nucleoprotein antibody (Custom Monoclonals International, Sacramento, CA, USA) at a dilution of 1:3000. A secondary biotinylated goat anti-mouse IgG antibody (DAKO Corp., Carpinteria, CA, USA) was applied to samples at a dilution of 1:250, followed by a signal amplification with avidin-biotin-conjugated peroxidase (elite Peroxidase, Vector Laboratories). Finally, the antigen-antibody complex was observed by its reaction with 3,3′-diaminobenzidine (DAB). Slides were counterstained lightly with hematoxylin and cover slipped.

Virus Isolation was performed from tissue collected at necropsy and frozen at −80 °C or refrigerated from 7 animals. Approximately 1 g of brain (*N* = 1) and or lung (*N* = 6) were placed in a stomacher bag with 10 mL cell minimal essential media (MEM), FBS and 20 mL of MEM 1x antibiotics. The mixture was processed in a stomacher-machine overnight. Fifteen milliliters of the mixture were added to a conical tube and centrifuged rpm for 20 min at 4 °C. From the supernatant, 0.45 uL were syringe filtered and inoculated into a 24-well plates of VeroDogSLAM cells (Ed Dubovi, Cornell) prepared 24 h previously. Assessment of cell cultures for cytopathic effect (CPE) occurred daily. Following one week without CPE, the virus isolation process was repeated, and numbers of wells doubled. Once CPE was noted after a maximal period of 2 weeks or two isolation attempts, direct immunofluorescence assay was performed. Direct immunofluorescence assay (FA) was performed on frozen sections of fresh tissue and on inoculated monolayer culture of Vero cells presenting cytopathic effect, followed by two serial passages, using a commercially available kit (VMRD Inc., Pullman, WA, Australia). For fresh-frozen tissue, submitted frozen brain or lung was mixed with OCT and imbedded in the cryostat −20 °C. Six-micrometer thick sections of OCT frozen tissue were cut and placed on positively charged slides (Probe-On Plus, Fisher Scientific, Springfield, NJ, Australia) and air dried overnight. Sections were then fixed with acetone at −20 °C for 20 min, air dried for approximately 30 min and stained with canine distemper virus conjugate (VMRD Inc., Pullman, WA, USA, Cat. #CJ-F-CDV-10 ML) for 30 min at 37 °C. Slides were washed with buffer solution (VMRD Inc., Pullman, WA, Australia) for 10 min and covered by VMRD FA melting fluid and coverslipped.

Reverse transcription PCR was performed from aliquots of frozen tissues collected at necropsy and virus isolates (for one tiger). Samples of lung (all Panthera, two raccoons and one skunk), brain (two raccoons and one skunk), kidney (one tiger) and lymph node (one raccoon) were thawed submerged in an excess of an RNA stabilizing solution (RNAlater-ICE, Ambion). Total RNA was extracted from tissues using TRIzol Plus RNA kit (TermoFisher Scientific, Waltham, MA, USA) per the manufacturer’s instructions. Reverse transcription of extracted RNA was performed using GenAmp RNA PCR Core kit (TermoFisher Scientific, Waltham, MA, USA) per manufacturer’s instructions. The entire H gene (1824 bp) was amplified from the frozen tissue via a commercially available PCR kit using previously published primers CDVff1, CDV-HS1, CDV-H for D, CDV-Hr2, and CDV-HS2 [53]. Amplified PCR products were purified using ExoSAP-IT kit (Affymetrix Inc, Santa Clara, CA, USA) and were directly sequenced at a commercial facility (University of Chicago Cancer Sequencing Facility). Global alignment with free-end gaps was performed on Geneious Prime. Sequence similarities were assessed based on percentage of sequence similarity/distance. Subsequently, sequences were mapped to an 1824bp cRNA linear hemagglutinin gene reference isolated from a free-ranging African lion (GenBank accession number JN812975.1).

### 4.3. Retrospective Epidemiologic Study

Epidemiologic assessment of the outbreak was performed through a questionnaire, on site interview between November 2015 and January 2016, mapping of the rescue facility with temporal distribution of diagnosed deaths between December 2015 and May 2016, and two follow-up interviews in June 2018. Epidemiologic data comprised animal age, sex, acquisition date, enclosure allocation around time of the outbreak, exposure to wildlife as well as vaccination and health history since arrival (when available). Statistical Fischer exact tests were performed with the online tool https://www.socscistatistics.com/tests/fisher/default2.aspx, accessed on 24 July 2018 for calculated odds ratio.

Sickness was defined as signs of respiratory and or gastrointestinal affliction noted by staff of the rescue center during the outbreak. Reported signs were nasal discharge, coughing, difficulty breathing and or orange diarrhea in one individual or group (Table 1).

## Figures and Tables

**Figure 1 pathogens-10-00544-f001:**
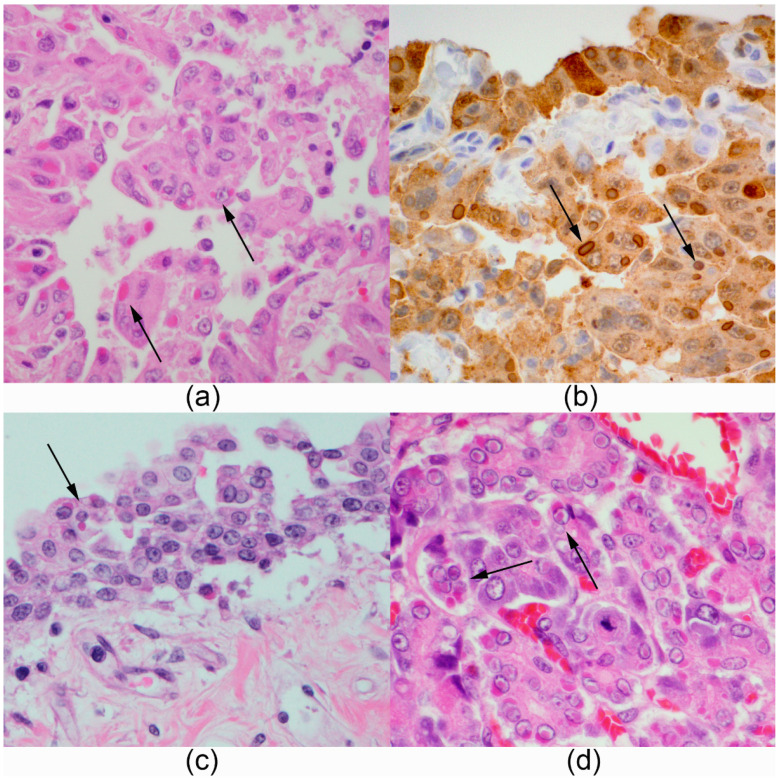
Canine distemper virus-typical intranuclear and intracytoplasmic, bright eosinophilic viral inclusions in multiple organs of *Panthera*: (**a**,**b**) Lion, lung with marked type II pneumocyte hyperplasia and syncytial cells containing viral inclusions (arrows, Hematoxylin and Eosin staining (H&E) and CDV-IHC); (**c**) Tiger #7, Renal pelvis, epithelial cells contain intracytoplasmic and intranuclear viral inclusions (arrows, H&E); (**d**) Tiger #7, pancreas, numerous exocrine epithelial cells contain viral inclusions (arrow, H&E).

**Figure 2 pathogens-10-00544-f002:**
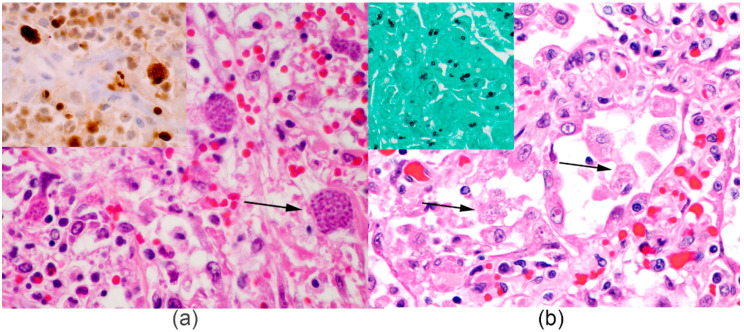
Opportunistic infections recorded in *Panthera*: (**a**, H&E) Tiger #5, lung, type II pneumocytes and macrophages are markedly enlarged (up to 70 μm in diameter) and filled with an intracytoplasmic parasitopohorus vacuole that contains numerous 2–4 μm, round to fusiform, basophilic tachyzoites. Inset is the IHC for *Toxoplasma gondii*, which confirms strong immunoreactivity within infected cells. (**b**, H&E) Tiger #6, lung, alveolar macrophages and few type II pneumocytes are enlarged and contain numerous intracytoplasmic round to ovoid, pale basophilic, 2–5 μm in diameter yeasts with 1–2 μm thick walls (arrows) that are also GMS-positive (black, inset). Microbiology culture was most compatible with *Histoplasma capsulatum*.

**Figure 3 pathogens-10-00544-f003:**
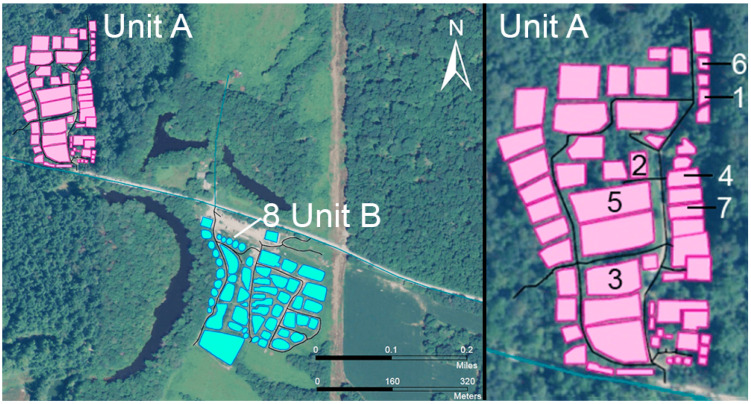
Schematic map of the rescue center, which was divided in two epidemiologic units A/Front (in pink) and B/Field (in turquoise). Inset has the order of the seven cases within unit A. Only one CDV-confirmed case was registered in Unit B.

**Table 1 pathogens-10-00544-t001:** Signalment and clinical signs reported (+) in *Panthera* confirmed to be infected with canine distemper virus post-mortem. Animals are listed chronologically from top to bottom.

Case	Animal	Sex	Age (Years)	Symptomatic Prior to Death (Days)	Lethargy	Anorexia	Nasal Discharge ^1^	Labored Breathing	Diarrhea ^2^	Neurologic
1	Tiger	Male	18	14	-	+	+	-	+	-
2	Tiger	Male	14	5	-	+	+	+	-	-
3	Lion	Male	20	<1	-	-	+	-	-	-
4	Tiger	Female	>16	3	+	+	-	-	-	-
5	Tiger	Male	18	33	-	-	+	-	+	-
6	Tiger	Male	>5	40	-	+	+	-	+	-
7	Tiger	Male	16	23	+	+	+	-	+	-
8	Tiger	Female	18	9	+	+	+	-	-	-

^1^ Mainly serous than mucopurulent nasal discharge. ^2^ Diarrhea was characterized by watery to pasty brown to orange feces, reported in multiple cages prior to and concomitantly to the death of the first two tigers.

**Table 2 pathogens-10-00544-t002:** Confirmatory tests performed per case and organs used. IHC: Immunohistochemistry; FA: Direct immunofluorescence assay; VI: Virus isolation; PCR: Polymerase chain reaction; H-gene: CDV H-gene sequencing. N.A. = Not attempted.

Case	FA-Lung	FA-Brain	Virus Isolation ^1^	IHC-Lung	IHC-Brain	PCR Lung	H-Gene
1	+	N.A.	-	+	N.A.	+	+
2	N.A.	N.A.	N.A.	+	N.A.	+	+
3	+	-	+	+	N.A.	+	+
4	+	N.A.	+	+	N.A.	+	+
5	+	N.A.	- ^2^	+	N.A.	+	+
6	+	N.A.	-	+	N.A.	+	+
7	-	-	-	+	N.A.	+	+
8	N.A.	N.A.	+	+	-	+	+

^1^ Performed only in lungs with exception of one case, case #4. ^2^ Homogenate of samples from lung and cerebrum were used.

**Table 3 pathogens-10-00544-t003:** Species and number of felids present in the rescue center during the outbreak are distributed based on their CDV-disease susceptibility, CDV-like clinical signs reported at time of outbreak and CDV-confirmed death. In brackets is the number of animals that had records confirming vaccination previous to the outbreak.

Species	Unit A	Unit B	CDV-Susceptible	CDV-Like Signs ^1^	CDV-Death
African Servals	6 (0)	0	n.r ^2^	0	0
Bobcat	9 (0)	2 (1)	11	0	0
Canada Lynx	1 (0)	0	1	0	0
Cougar	5 (0)	2 (2)	n.r	0	0
Leopard	3 (0)	9 (4)	12	0	0
Lion	8 (1)	13 (7)	21	2	1
Savana cat	5 (0)	0	n.r	0	0
Tiger	47 (4)	76 (16)	123	23 (1)	7 (1)
Geoffrey’s cat	1 (0)	0	n.r	0	0
Ocelot	0	1 (0)	n.r	0	0
Total	85 (5)	103 (28)	168	25 (1)	8 (1)

^1^ as described in Table 1. ^2^ n.r. = no scientific reports of CDV-disease in these species at time of writing.

**Table 4 pathogens-10-00544-t004:** Comparison of the base sequence as well as amino acid variability at residues 178, 519, 530 and 549 of the H-gene sequences harvested from sympatric Panthera (N = 8) and raccoon (N = 1) necropsied during the outbreak and from wildlife necropsied shortly thereafter (N = 4). Animals are listed chronologically. R = Arginine, I–Isoleucine, G = Glycine, H= Histidine.

Sample Origin	Residue 178	Residue 519	Residue 530	Residue 549
lung, case 1 ^1^	GGC-G	AGA-R	GGT-G	CAC-H
lung, case 2	GGC-G	AGA-R	GGT-G	CAC-H
lung, case 3	GGC-G	AGA-R	GGT-G	CAC-H
lung, case 4	GGC-G	AGA-R	GGT-G	CAC-H
lung, case 5	GGC-G	AGA-R	GGT-G	CAC-H
lung, case 6	GGC-G	AGA-R	GGT-G	CAC-H
lymph node, raccoon IN	GGC-G	AGA-R	GGT-G	n.a. ^3^
lung, case 7	GGC-G	AGA-R	GGT-G	CAC-H
virus isolate, case 8	GGC-G	AGA-R	GGT-G	CAC-H
lung, skunk IL 1 ^2^	GGC-G	ATA-I	GGC-G	CAC-H
lung, skunk2 IL ^2^	GGC-G	ATA-I	GGC-G	CAC-H
brain, racoon IL ^2^	GGC-G	AGA-R	GGT-G	CAC-H

^1^ See Table 1 for case detail. ^2^ These animals were necropsied between 11th April and 3rd May 2016 and had CDV-consistent lesions in both brain and (lesser) lung. ^3^ This sequence was 1608 bp long and did not include residue 549.

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
