# Peer review of "Sylvatic Canine Morbillivirus in Captive Panthera Highlights Viral Promiscuity and the Need for Better Prevention Strategies"

_pathogens, 2021, doi:10.3390/pathogens10050544_

Round 1

Reviewer 1 Report

Linhares and colleagues describe a case report of a CDV outbreak in an exotic felid rescue centre. Seven tigers and one lion succumbed to the infection, with signs of pneumonia and opportunistic infections. The virus was phylogenetically similar to CDV detected in raccoons and skunk, suggesting spillover infection. In large felids the risk of severe disease and death was increased in animals sharing a fence or enclosure with other infected animals, demonstrating felid-to-felid transmission.

This is a useful outbreak description, but my major comment is that the authors should come to another main conclusion in the title, abstract and discussion. Instead of emphasizing that the outbreak highlights promiscuity of CDV, I believe the outbreak shows (again) that large felids are highly susceptible to CDV, and prophylactic vaccination (if possible combined with surveillance of neutralizing antibody titres) is crucial. This outbreak emphasizes the necessity of a CDV vaccine that is safe and effective in large felids.

Specific comments:

  1. Title: the combination of sylvatic and spillover seems repetitive.
  2. Title: Rather than these observations highlighting viral promiscuity, I would suggest changing the title to conclude that these observations highlight the importance of distemper vaccination in felid rescue centres and zoos (see also main comment above).
  3. Line 21: change ‘lead’ to ‘led’.
  4. Line 22: change ‘primarily’ to ‘primary’.
  5. Line 23: change ‘particles’ to ‘antigens’.
  6. Line 35: write ‘distemper virus’ without capitals.
  7. Line 40: change ‘through aerosolized virions’ to ‘via the respiratory tract’. Since the majority of CDV host species are predators, the virus can also be transmitted through large droplets, fomites and even by blood when a naïve animal predates on an infected animal.
  8. Line 52: the authors leave out the most important aspect of the pathogenesis of distemper, namely the fact that the virus causes a severe immunosuppression. CDV infects a high percentage of B- and T-lymphocytes, both in peripheral blood and in lymphoid tissues. The infection percentages are much higher than those seen in measles, and so is the level and duration of lymphopenia and the resulting level of immunosuppression. As a consequence, the adaptive immune response to other pathogens is largely ablated. See e.g. Beineke et al Vet Immunol Immunopathol 2009 (listed as ref 13 in the manuscript)
  9. Line 85: change ‘relative’ to ‘relatively’.
  10. Line 101: change ‘Canine Distemper Virus’ to ‘CDV’.
  11. Line 208/209: it is unclear to me what the authors mean to say with this sentence. I propose to delete it.
  12. Line 220-226: this table has no added value to the manuscript, and can be removed.
  13. Line 253: skunk are members of the family Mephitidae, and do not belong to the Mustelidae (although they used to).
  14. Line 302: this sentence seems to be incomplete.
  15. Line 312: change ‘particle’ to ‘antigen’.
  16. Line 303-316: the authors should discuss the possibilities (or challenges) to collect blood samples from large felids. Serology (preferable CDV neutralization) would be the most appropriate method to validate the efficacy of vaccination (as briefly mentioned in lines 315-316, but it is not explained if this was done or even possible).
  17. Lines 371-387: CDV sequences generated in this study should be submitted to GenBank, and accession numbers should be provided in the manuscript.
  18. Reference 4: correct typo in journal name.
  19. Reference 10: remove ‘editor’ at the end?
  20. Reference 12: correct journal name.
  21. Reference 25: correct question marks in author name.

Reviewer 2 Report

This paper is interesting with broad impact and the authors to a good job of presenting the data. There are a few details within the manuscript that if clarified, can potentially broaden the impact of this publication. Since the clinical presentation of Panthera with sylvatic canine morbillivirus is not well described in the literature, it would be of additional interest to include more information on the clinical presentation and course. For example, the authors mention that several of the Panthera in this report were euthanized, but ~12 cases had CDV-like disease and recovered. How did those cases differ (age? vaccination? supportive care? comorbidity?). Since immunization is mentioned in the title of the manuscript, inclusion of additional details on the vaccination of these felids would also strengthen the publication.

The authors connection among the IL and IN raccoons, rescue animals, and the snow leopard case are not clear to me. The authors describe that these viruses are genetically similar, but it is unclear how this is relevant to the outbreak. I wonder if more details on the proximity (actual mileage/kilometer distance and expected virus mutation relative to that distance might be helpful. If the same virus occurred in neighboring states, but the locations are within the expected range of a single raccoon, then these cases might represent one outbreak/spillover event. As currently described in this paper, I am left wondering if sylvatic canine morbillivirus in this particular region, or this particular morbillivirus, is more likely to have spillover into Panthera than other mutations of CDV or if these are just random cases with poor vaccination protection.

There have been several papers discussing vaccination for CDV among zoo housed species or at risk species near wildlife published recently.  I wonder if the authors can give suggested guidelines for when known morbillivirus is occurring within neighboring wildlife. Should veterinarians increase vaccination protocols for morbillivirus if there are known raccoon/skunk disease? Should zoos/rescue centers conduct surveillance of wildlife? Should wildlife trapping (euthanize versus trap and move) increase? Are there other mitigation practices that should occur? Did this rescue center implement practices to prevent recurrence?

The authors use several methods to confirm CDV among these Panthera cases. Could details on CDV-FA, RT-PCR, H-Gene, VI, IHC, FA be included? Perhaps explaining why some of these would be negative but other testing positive?

Authors seem to use the terms “cat”, “animals”, andfelids” interchangeably.  This is somewhat confusing since “animals” could include Panthera or raccoons. Cat is more commonly used to refer to domestic cats, not Panthera species. Could the authors use more detailed language to specify what species they are discussing?

Table 1. Odd spacing in table. Please include info on how were these tigers/lino diagnosed? Antemortem or post-mortem CDV diagnosis?

Line 126 – Any chance you have the distance in miles or kilometers? Curious to what “nearby” means?

Line 151 – What is the distance separating Unit A and Unit B? Shared staffing? Any recent felids added to Unit A or B. Do they move felids between Unit A or B?

Line 150 & Table 3– This part is somewhat confusing. In the earlier part, the discussion is only about necropsied animals, but here you present data on animals with “CDV-like illness” can you define this? Or perhaps clarify this part? What is the number within the column of CDV- susceptible? Are these the number of publications where these species have been reported to be susceptible to CDV? Can you put in formation on type of vaccination and when vaccine was given in this table? There were only 4 animals vaccinated, were these single or multiple vaccinations for these four animals, recombinant or live virus vaccines? Perhaps a separate table of vaccination history would be helpful.  

Figure 3 – The black writing is hard to see within the figure. Can you put a line or some sort of more clear delineation between the inset and zoomed out landscapes?

Line 182 – “These findings suggest spread among cats after initial spillover from wildlife”. This sentence may be better in discussion. Needs to be explained somewhat. Could an infected felid have been added to collection and started the outbreak? Could staff have contributed to transmission? Could a diet item have contributed to transmission?

Lines 185 – 200 – This portion of the paragraph is confusing. If all felids were vaccinated around time of rescue, then why were only 33 animals known to have record of CDV vaccination? Can you include the viruses that are covered by Fel-O-Vax IV + Calici Vax, Nobivac Puppy-DPv? PUREVAX Ferret Distemper (Merial)? Should we expect Fel-O-Vax to protect against sylvatic CDV?

206 – What vaccine was used to vaccinate during the outbreak?

205 – Was this case with non-CDV related death really unrelated? What was the underlying cause of pneumonia and meningitis? What are the rates of false negative among VI, IMHC, IFA? Can we assume this individual was Panthera? Could this have been a vaccine reaction? Could this have been death from one of the other MLV within a combo vaccine?

Table 4. It looks like there are IN and IL raccoons tested, but the table indicates raccoon (N=1). Were other dead wildlife found around the rescue center? Seems odd so many felids were affected but only one dead raccoon. Any chance this organism came in from an infected newly added felid, spread, then spilled over into wildlife?

Line 293 – It sounds like the vaccination implementation may have shifted over the past decade at the rescue center. Also wonder about immunocompetence of felids that got sick and died, seems most were quite aged.

Line 299 – Typo? Should “order” be “other”?

Line 302 – Sentence fragment? Unsure of authors meaning of this sentence.

Line 306 – Need to include the vaccination records of the sick and dead felids.
